# A predictive molecular signature consisting of lncRNAs associated with cellular senescence for the prognosis of lung adenocarcinoma

**Anbang Liu[☯], Xiaohuai Wang[☯], Liu Hu, Dongqing Yan, Yin Yin, Hongjie Zheng, Gengqiu Liu, Junhang Zhang\*, Yun Li[ID]\***

Department of Thoracic Surgery, The Seventh Affiliated Hospital, Sun Yat-sen University, Shenzhen, China

☯ These authors contributed equally to this work.
\* liyun28@mail.sysu.edu.cn (YL); zhangjh33@mail.sysu.edu.cn (JZ)

**Data Availability Statement:** About TCGA database, URLs: https://portal.gdc.cancer.gov. DOIs: Grossman, Robert L., Heath, Allison P., Ferretti, Vincent, Varmus, Harold E., Lowy, Douglas

## Abstract

The role of long noncoding RNAs (lncRNAs) has been verified by more and more researches in recent years. However, there are few reports on cellular senescence-associated lncRNAs in lung adenocarcinoma (LUAD). Therefore, to explore the prognostic effect of lncRNAs in LUAD, 279 cellular senescence-related genes, survival information and clinicopathologic parameters were derived from the CellAge database and The Cancer Genome Atlas (TCGA) database. Then, we constructed a novel cellular senescence-associated lncRNAs predictive signature (CS-ALPS) consisting of 6 lncRNAS (AC026355.1, AL365181.2, AF131215.5, C20orf197, GAS6-AS1, GSEC). According to the median of the risk score, 480 samples were divided into high-risk and low-risk groups. Furthermore, the clinicopathological and biological functions, immune characteristics and common drug sensitivity were analyzed between two risk groups. In conclusion, the CS-ALPS can independently forecast the prognosis of LUAD, which reveals the potential molecular mechanism of cellular senescence-associated lncRNAs, and provides appropriate strategies for the clinical treatment of patients with LUAD.

## Introduction

Globally, among all cancers, lung cancer ranks first in mortality and second in incidence nowadays [1]. Generally, there are two pathological types named non-small cell lung cancer (NSCLC) and small cell lung cancer. Lung adenocarcinoma (LUAD) is the most prevalent histological subtype of primary NSCLC, comprising roughly of 50% of cases [2]. According to the different stages of LUAD, patients need to receive comprehensive treatments including surgery, chemoradiotherapy, targeted therapy and immunotherapy, but the prognosis of advanced LUAD is still poor. Therefore, LUAD remains a serious public health problem that requires continuous attention, and it is indispensable to discover more drug targets, biomarkers and deeper understanding of other molecular and biochemical factors to contribute to a better prognosis of patients.

Cellular senescence is characterized by a permanent cessation of cell proliferation. The occurrence of cellular senescence is associated with many different triggers, such as DNA damage and release of inflammatory cytokines, etc. Increasing evidence shows that many aging-related diseases can be prevented or delayed, including cancer, by regulating the biological

R., Kibbe, Warren A., Staudt, Louis M. (2016) Toward a Shared Vision for Cancer Genomic Data. New England Journal of Medicine375:12, 1109-1112. About CellAge database, URLs: https://genomics.senescence.info/cells/. DOIs: Avelar, R. A., Ortega, J. G., Tacutu, R., Tyler, E. J., Bennett, D., Binetti, P., Budovsky, A., Chatsirisupachai, K., Johnson, E., Murray, A., Shields, S., Tejada-Martinez, D., Thornton, D., Fraifeld, V. E., Bishop, C. L., & de Magalhaes, J. P. (2020) "A multidimensional systems biology analysis of cellular senescence in aging and disease." Genome Biology 21(1):91.

**Funding:** The authors received no specific funding for this work.

**Competing interests:** The authors have declared that no competing interests exist.

process of cellular senescence [3,4]. A large number of researches suggest that cellular senescence profoundly regulates tumor microenvironment by promoting the infiltration of some immune cells, activating many signaling pathways and releasing inflammatory cytokines [5–7]. Xue et al. found that in an in vivo model of hepatocellular carcinoma, restoration of *p53* could induce senescence of liver tumor cells and promote elimination of these cells by the innate immune system [8]. Some studies have found that *eIF3A R803K* mutation promotes resistance to chemotherapy by inducing tumor cell senescence and the higher expression of cellular senescence genes in human hepatocellular carcinoma is close to the bad prognostic outcome in terms of overall survival (OS) [9,10]. Furthermore, senescent thyroid cells can induce neighboring non-senescent thyroid cancer cells to invade and proliferate. The xenograft model also shows that senescent thyroid cells and cancer cells co-implanted increased lymphovascular invasion and lymph node metastasis through the CXCL12/CXCR4 axis [11].

Long noncoding RNA (LncRNA) refers to a kind of non-coding RNAs that are longer than 200nt and cannot encode proteins. LncRNA can regulate coding genes through chromatin remodeling, transcriptional control and post-transcriptional processing [12]. LncRNA mainly has four functions. The first function is "signaling function", which means the transcript of lncRNA itself serves as the marker of biological events including allelic specificity, structural specificity and various stimuli. Secondly, it is so-called "molecular decoy" that lncRNA can bind with proteins and miRNAs, etc. The third is "guiding". LncRNA acts in cis or trans to target the bound effector molecule to the position of specific DNA sequence. Finally, lncRNA acts as a "molecular scaffold" can take part in the assembly of transcriptional regulatory complexes [13]. On the basis of previous researches, it has been discovered that lncRNAs with aberrant expression contribute substantially to the onset and development of numerous diseases, tumor growth and metastasis are particularly influenced [14]. Several lncRNAs have the potential to be tumor biomarkers and therapeutic targets for the detection and treatment of cancer [15]. However, there are few reports on cellular senescence-associated lncRNAs in terms of LUAD.

Through this research, we constructed a cellular senescence-associated lncRNAs predictive signature (CS-ALPS), and evaluated its prognostic value in LUAD patients. Furthermore, we further performed immune-related analyses and sensitivity analyses of common drugs to explore a more significant clinical treatment plan.

## Materials and methods

### Acquisition of LUAD dataset and cellular senescence- related gene set

Fig 1 depicts a flow diagram of our work. The Cancer Genome Atlas (TCGA) database (https://portal.gdc.cancer.gov/) was accessible to download the RNA sequencing (RNA-seq) dataset, survival information, and associated clinicopathologic parameters of LUAD. Meanwhile, we obtained 279 cellular senescence-related genes from the CellAge database (https://genomics.senescence.info/cells/) (S1 Table).

### Enrichment analyses of differentially expressed genes related to cellular senescence

We extracted the expression matrix of genes relevant to cellular senescence from the RNA-seq dataset, and used the "limma" package to draw the volcano plot of differentially expressed genes (DEGs). We considered the false discovery rate (FDR) < 0.05 and | log2 fold change (FC) | > 1 as the criteria of significantly differential expression, and further established a protein-protein interaction (PPI) network about DEGs through the STRING database (https://cn.string-db.org/). Then, we visualized the network by Cytoscape software, and screened out hub

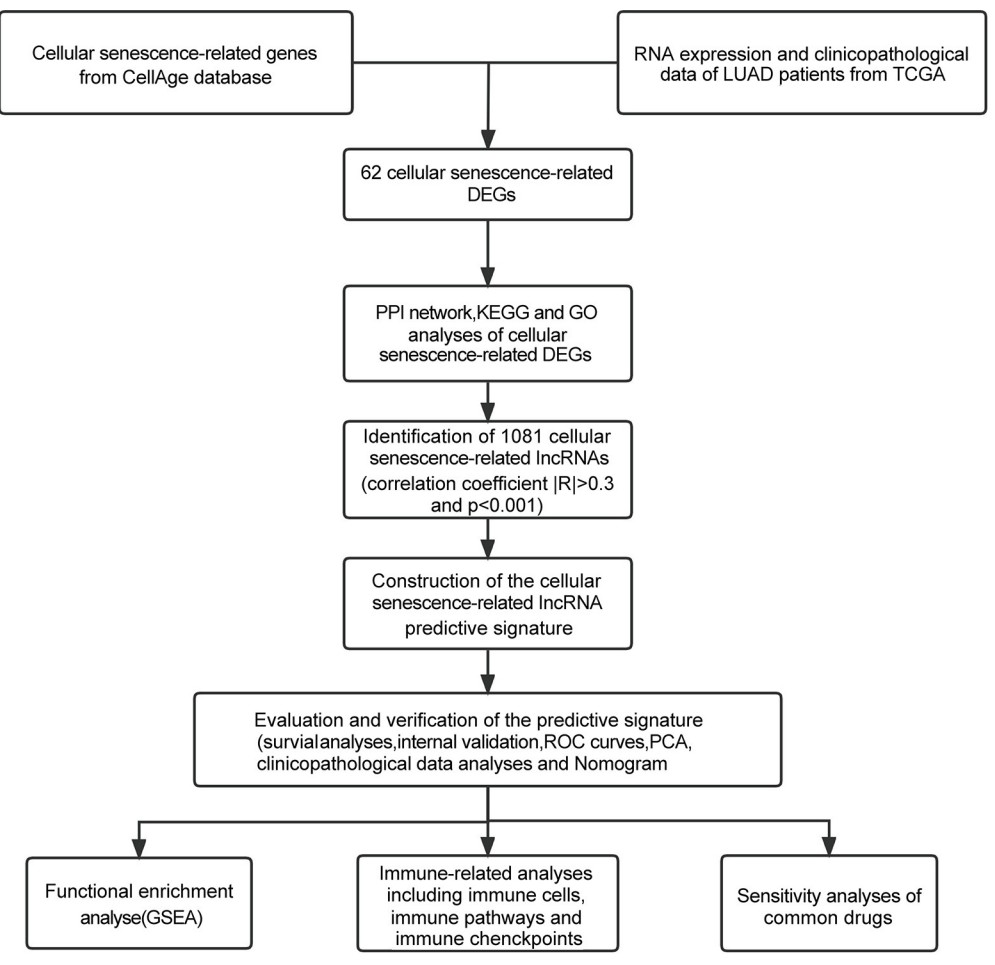

**Fig 1. The flow diagram of our research.**

genes based on degree values by the application "cytoHubba". The functions of DEGs were excavated by Gene Ontology (GO) and Kyoto Encyclopedia of Genes and Genomes (KEGG) analyses using the R package "clusterProfiler" and "GOplot" [16,17].

## Construction, evaluation and verification of the predictive signature

We calculated the correlation between DEGs and lncRNAs and regarded the correlation coefficient | R | > 0.3 and p < 0.001 as screening standard. In order to investigate the connection between cellular senescence-associated lncRNAs and the OS, the univariate Cox regression analysis was initially performed. Least absolute shrinkage and selection operator (LASSO) was further used to obtain the cellular senescence-associated lncRNAs through the "glmnet" package and taking 10-fold cross-validation [18,19]. Lastly, we used the multivariate Cox regression analysis to identify and construct an efficient CS-ALPS on the basis of the following computational formula:

$$\text{Risk score} = \sum_{i=1}^{n} \text{Coef}(i) \times \text{Expr}(i)$$

Coef(i) and Expr(i) represent the multivariate Cox regression coefficient and corresponding lncRNA expression level, respectively. We could calculate the risk score for each LUAD patient

using the formula. Furthermore, on the basis of the median risk score, we divided LUAD patients into two categories, named respectively high-risk and low-risk groups.

We tested the reliability of the CS-ALPS by principal component analysis (PCA). Additionally, through the "rms" package, we made a nomogram for the purpose of predicting LUAD survival at 1-year, 3-year, and 5-year. The performance of the nomogram was evaluated using calibration curve.

Gene set enrichment analysis (GSEA) 4.1.0 could help to investigate which pathways were mostly enriched in high-risk and low-risk groups, respectively [20]. Normalized enrichment score | NES | ≥ 1 and nominal p <0.05 were taken into account as our standard for statistical significance.

### Immune-related analyses of the CS-ALPS

Immune-related analysis was performed for the CS-ALPS, including the infiltration of stromal and immune cell. We calculated the StromalScore, ImmuneScore, and ESTIMATEScore respectively by the R package "estimate" and the Wilcoxon signed-rank test was applied to compare immune scores between the different risk groups [21]. Following that, single-sample gene set enrichment analysis (ssGSEA) was used to compute the infiltration scores of immune cells and explore the biological functions of immune-related pathways between high- and low-risk groups through the "GSVA" and "GSEABase" packages [22].

### Sensitivity analysis of common drugs

To explore whether the CS-ALPS was a guidance for LUAD therapy, we chose the half-maximal inhibitory concentration (IC50) as a reference to evaluate the predictive role of the signature in chemotherapy and targeted therapy. IC50 of some common drugs for chemotherapy and targeted therapy of LUAD were calculated by the R package "pRRophetic" [23].

### Statistical analysis

R software (Version 4.0.5) was carried out conducting all statistical analyses. If not otherwise stated, p < 0.05 was thought to be statistically meaningful. The Wilcoxon signed-rank test was utilized to compare differences between two groups. The Kruskal–Wallis test was employed to compare differences among three or more groups. The Kaplan–Meier method was employed for survival analyses by the R package "survminer"and "survival". We used the "timeROC" and "survivalROC" package to generate the time-dependent and conventional ROC curves. According to the area under the curves (AUCs), the predictive efficacy could be determined.

## Results

### Identification and analyses of the cellular senescence-related DEGs

We obtained 62 cellular senescence-related DEGs, comprising 39 upregulated and 23 downregulated genes (Fig 2A, S2 Table). According to the visual PPI network, we discovered the five most valuable genes as the hub genes based on their degree values (Fig 2B). As is shown in KEGG analyses, the enrichment of DEGs was most prominent along these pathways, such as cellular senescence, cell cycle, the p53 signaling pathway, oocyte meiosis, bladder cancer, cushing syndrome, endocrine resistance and progesterone–mediated oocyte maturation (Fig 2C). Then, GO analyses revealed that these DEGs involved in the cell aging, cellular response to chemical stress, the replicative senescence, negative regulation of mitotic cell cycle, regulation of gliogenesis and so on (Fig 2D).

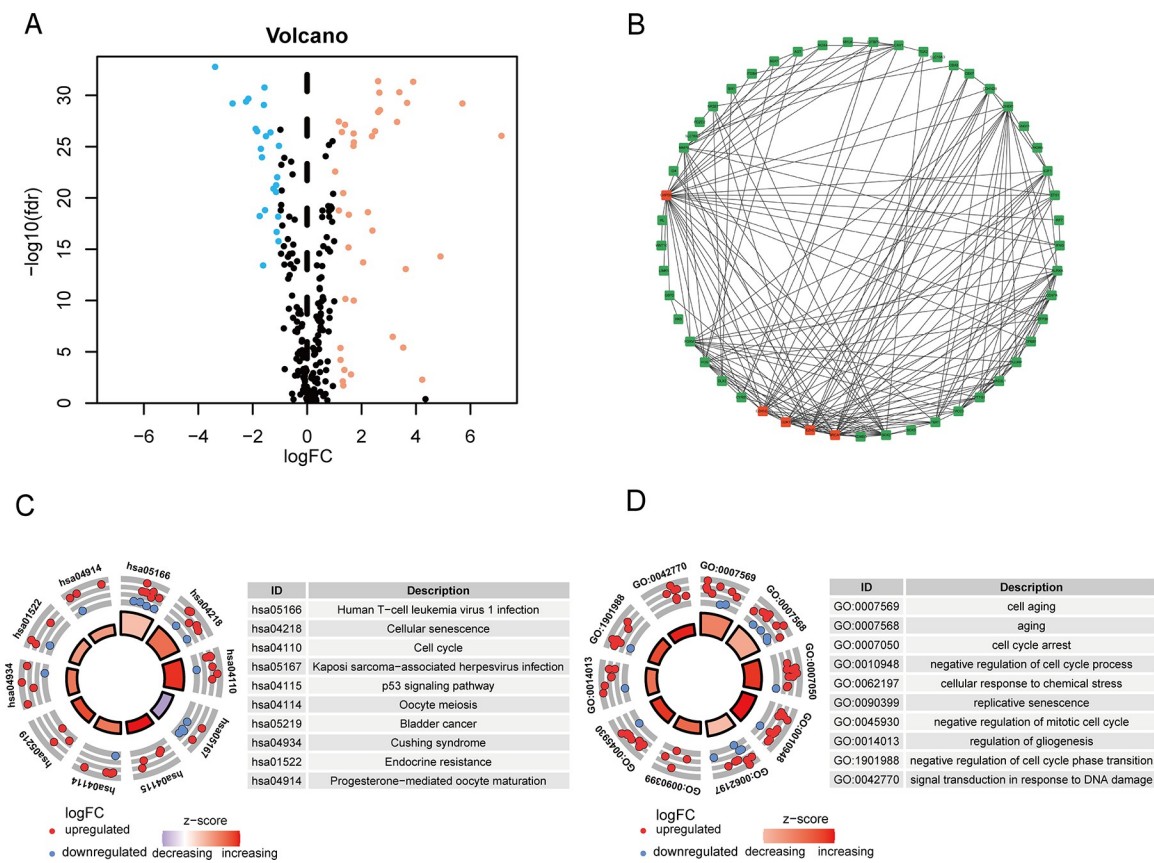

**Fig 2. Enrichment analyses.** (**A**) The volcano plot of 62 cellular senescence-related DEGs. (**B**) The PPI network of cellular senescence-related DEGs, red squares represent top 5 hub genes. (**C**, **D**) KEGG and GO analysis of cellular senescence-related DEGs, respectively.

## Identification and internal validation of the CS-ALPS to predict LUAD prognosis

We screened out 1081 cellular senescence-associated lncRNAs (S3 Table). Moreover, the 480 LUAD patients including both RNA expression data and survival data were acquired from TCGA dataset, and the OS of these samples was more than 30 days old (S4 Table). 66 lncRNAs were connected to the prognosis of LUAD following the outcomes of the univariate Cox regression analysis (S1 Fig, S5 Table). Then, LASSO identified that 29 lncRNAs were involved in cellular senescence and the multivariate Cox regression analysis identified 6 cellular senescence-associated lncRNAs (*AC026355.1*, *AL365181.2*, *AF131215.5*, *C20orf197*, *GAS6-AS1*, *GSEC*) (S6 and S7 Tables, S1 File). Finally, we constructed the CS-ALPS according the above analysis. The expression value of 6 cellular senescence-associated lncRNAs was displayed as a heatmap (Fig 3A). Fig 3B showed the correlation between 6 cellular senescence-associated lncRNAs and 12 co-expressed mRNAs. The co-expression network of 15 pairs of lncRNA-mRNA was shown in Fig 3C (| R | > 0.3 and p < 0.001). Among these pairs, *GSEC* was found to be co-expressed with 5 cellular senescence-related genes (*IGFBP3*, *CHEK1*, *G6PD*, *GAPDH* and *ASPH*). *AF131215.5* had co-expressed relationship with 4 cellular senescence-related genes (*ETS1*, *NTN4*, *PDZD2* and *SLC16A7*). *GAS6-AS1* had co-expressed relationship with *CBX7* and *SLC16A7*. *AL513550.1* had co-expressed relationship with *KDM5B* and *SLC16A7*. *AL365181.2* was co-expressed with *G6PD*. *C20orf197* was co-expressed with *WNT16*. We also found that *AL365181.2* and *GSEC* were risk factors, however, the other four lncRNAs were

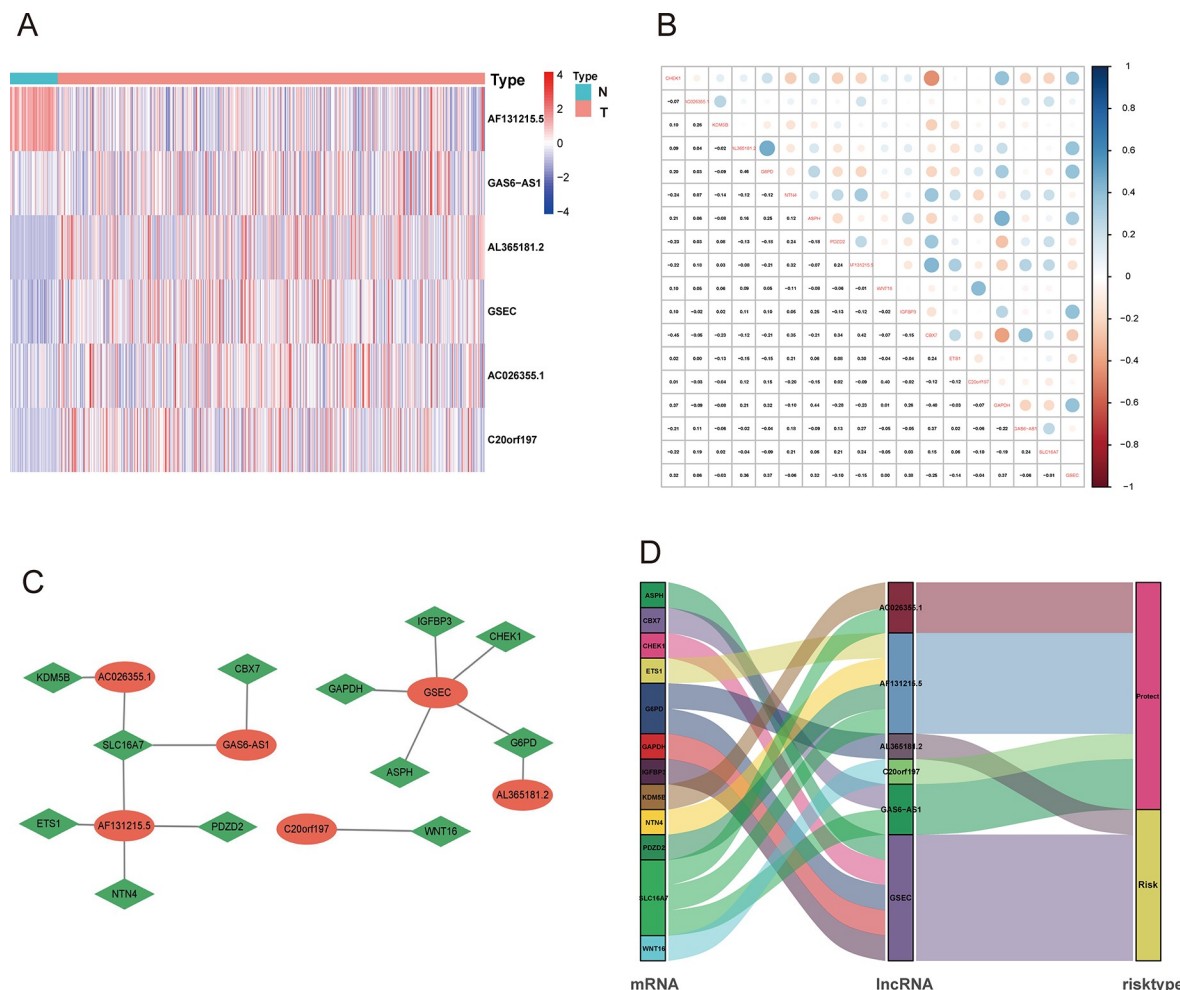

**Fig 3. Construction of the CS-ALPS.** (**A**) The heatmap of 6 cellular senescence- associated lncRNAs in tumor and normal tissues. (**B**) The correlation analysis between 6 cellular senescence-associated lncRNAs and 12 co-expressed mRNAs. (**C**) The co-expression network of lncRNAs and mRNAs. (**D**) The Sankey diagram for cellular senescence-associated lncRNAs, mRNAs and risk status. T, tumor; N, normal.

protective factors (*AC026355.1*, *AF131215.5*, *C20orf197*, *GAS6-AS1*) (Fig 3D). We designed a formula used for risk score. Risk score = (-0.270×*AC026355.1* expression) + (0.231×*AL365181.2* expression) + (-0.616×*AF131215.5* expression) + (-0.580×*C20orf197* expression) + (-0.627×*GAS6-AS1* expression) + (0.505×*GSEC* expression).

We obtained the median after determining the risk score for each sample using the formula. Subsequently, we divided the samples into high- and low-risk groups based on the median. The risk scores of two groups were shown in Fig 4A. As the scores climbed, so did the likelihood of death (Fig 4B). Compared with those at high risk, the OS of low-risk patients was noticeably longer (Fig 4C, p = 9.925e−10). The AUCs of 1-year, 3-year, and 5-year survival were respectively 0.745, 0.676 and 0.72, which manifested that the signature had relatively good predictive performance (Fig 4F). In order to verify the applicability of CS-ALPS, afterwards, two internal validation cohorts were formed by randomly dividing the LUAD patients. The results of the OS were consistent with the total cohort of predictive signature (Fig 4D, p = 1.042e−05; Fig 4E, p = 3.293e−05). Meanwhile, the two validation cohorts both showed similar predictive performance to the total cohort according to the ROC curves. The AUCs of 1-year, 3-year and 5-year survival were respectively 0.728, 0.671 and 0.72 in the first cohort

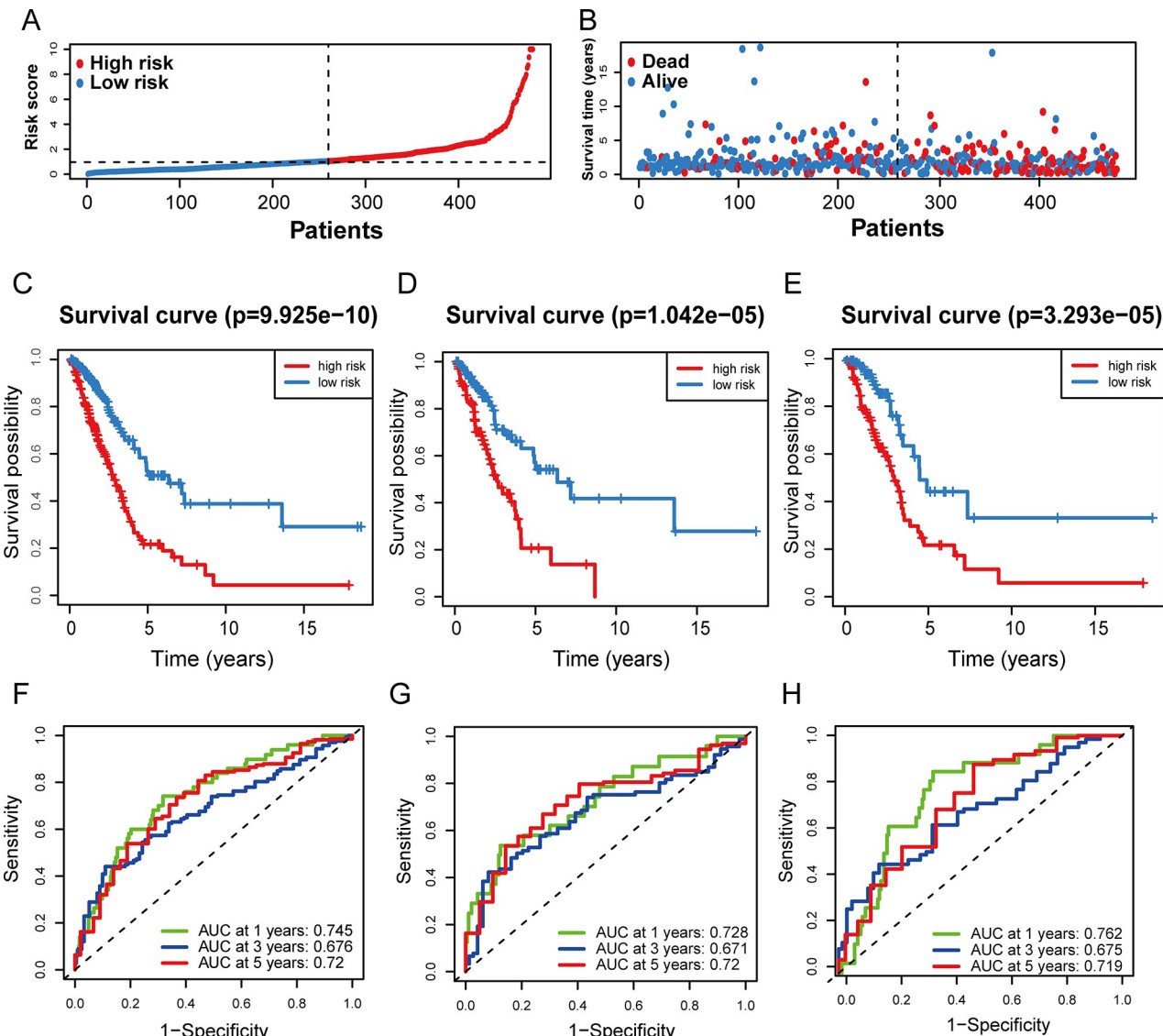

**Fig 4. Correlation between the CS-ALPS and LUAD prognosis. (A)** The distribution of LUAD patients. **(B)** A scatter plot based on each sample's OS and risk score. **(C-E)** The survival curves of LUAD patients in the total cohort and two internal cohorts, respectively. **(F-H)** The ROC curves for the total cohort and the two internal cohorts at 1-year, 3-year, and 5-year, respectively.

(Fig 4G) and those of the second cohort were respectively 0.762, 0.675 and 0.719 (Fig 4H). PCA 3D plots were used to distribute LUAD patients based on the entire genome, gene sets related to cellular senescence, cellular senescence-associated lncRNAs, and the CS-ALPS (Fig 5A–5D). The result identified that the CS-ALPS obviously discriminated degree of risk in different quadrants. On the contrary, the other groups could not be effectively distinguished, further supporting the reliability of the CS-ALPS.

## The CS-ALPS is an independent prognostic indicator in LUAD patients

The univariate Cox regression analysis revealed that stage, T, N and the CS-ALPS were closely connected to the OS of LUAD (Fig 6A). Stage and the CS-ALPS were identified as two independent predictors after multivariate Cox regression analysis (Fig 6B). Fig 6C indicated that

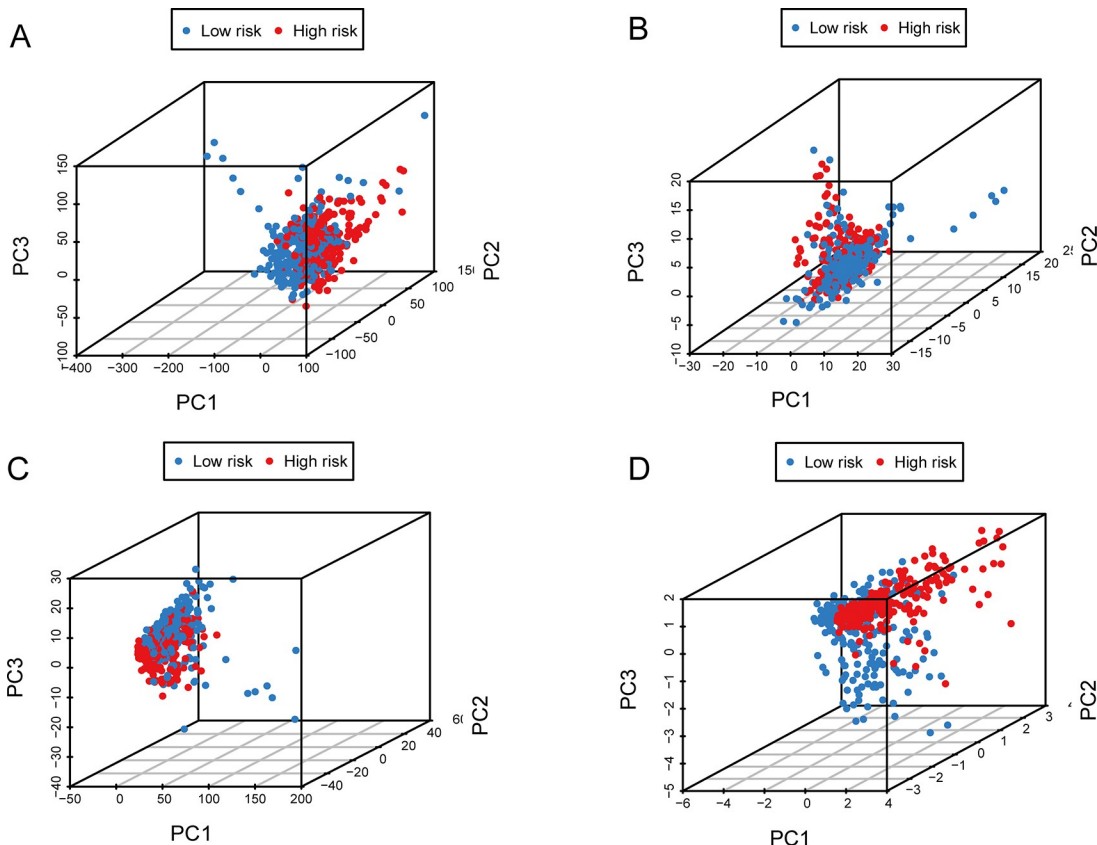

**Fig 5.** PCA 3D plots based on (A) the entire genome, (B) the gene sets related to cellular senescence, (C) the cellular senescence-associated lncRNAs, (D) the CS-ALPS.

the best predictor among all factors was the risk score in LUAD because of the largest AUC (AUC = 0.687). The differences of all clinicopathological factors were further studied, and the result revealed that N, T, stage and survival state were statistically different from the other factors between the two risk groups (Fig 6D). Next, we obtained the data of 328 LUAD patients including all expression data, survival data and clinicopathological information to investigate the relationship between the CS-ALPS and patients' prognosis. These samples were divided into different clinical subgroups on the basis of different clinicopathological factors. The results indicated that CS-ALPS as a risk factor was a relatively better predictor for LUAD in different clinicopathological subgroups except M1 subgroup (Fig 6E).

## The nomogram can predict the OS of LUAD

We further constructed a nomogram including stage and the CS-ALPS, which was able to forecast LUAD survival for 1-year, 3-year, and 5-year (Fig 7A). For the sake of evaluating the predictive efficacy of the signature, the calibration curves were performed. The findings showed that the predictive survival rates were fairly consistent with the actual survival rates at 1-year, 3-year, and 5-year (Fig 7B–7D).

## Identification of CS-ALPS associated pathways to explore biological functions

Enrichment analysis of the CS-ALPS was further carried out by GSEA to probe the underlying functions between two risk groups (Fig 8). The results indicated that the high-risk patients

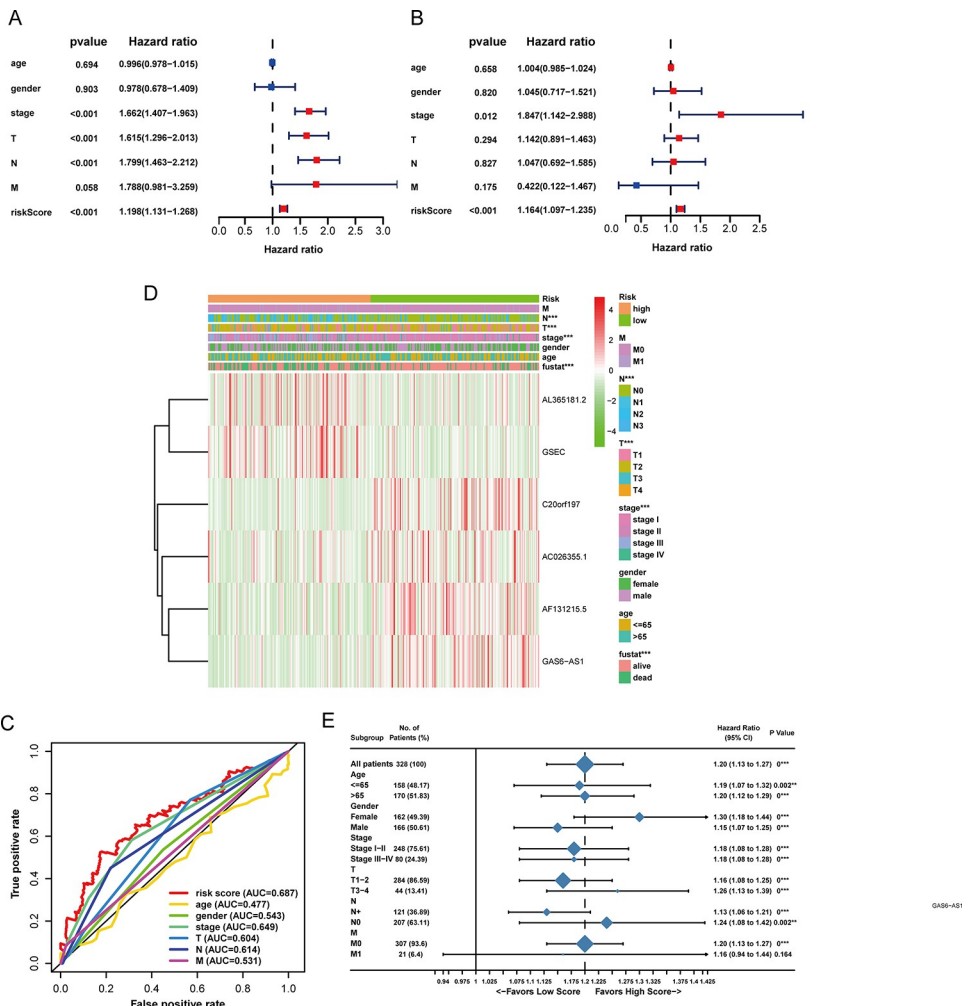

**Fig 6. Relationship between the CS-ALPS and clinicopathological factors.** Forest plots for (**A**, **B**) The univariate and multivariate Cox regression analyses of the CS-ALPS and clinicopathological factors. (**C**) The ROC curve of the CS-ALPS and clinicopathological factors. (**D**) Distribution heatmap of 6 lncRNAs and clinicopathological factors. (**E**) A forest plot for clinicopathological subgroup analysis. T, tumor; N, lymph node; M, metastasis. *p < 0.05; **p < 0.01; ***p < 0.001.

were connected to tumor-associated pathways, such as DNA replication (NES = 2.03 and nominal p <0.001), RNA degradation (NES = 1.81 and nominal p = 0.008), p53 signaling pathway (NES = 1.69 and nominal p = 0.013) and so on. The results also revealed that the low-risk group was closely associated with tumor-immune related pathways, such as FC epsilon RI signaling pathway (NES = -1.86 and nominal p = 0.004), mTOR signaling pathway (NES = -1.63 and nominal p = 0.006) and JAK/STAT signaling pathway (NES = -1.62 and nominal p = 0.031). The above results fully proved that cellular senescence-associated lncRNAs were linked to tumor immune microenvironment (TIME) in our predictive signature.

## The profile of immune infiltration between high- and low-risk groups

The three violin plots respectively showed StromalScore, ImmuneScore and ESTIMATEScore in two risk groups. Notably, the three types of scores were all significantly higher in low-risk patients (Fig 9A–9C). To further investigate the association between the CS-ALPS and

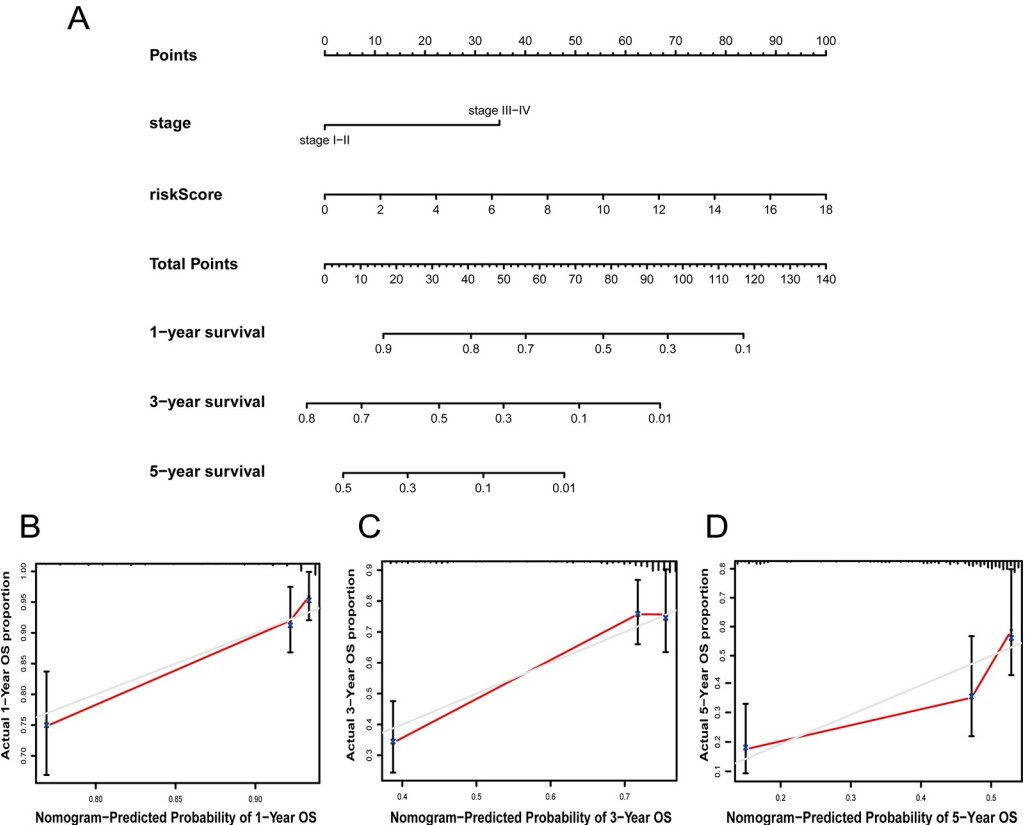

**Fig 7. Nomogram and calibration.** (**A**) A nomogram of predicting LUAD survival at 1-year, 3-year, and 5-year. (**B-D**) The calibration curves for predictive accuracy of the CS-ALPS.

immune functions, we assessed the ssGSEA enrichment scores for several immune cell subgroups and immune-related pathways. The analysis showed that most of the immune cell infiltration happened in the low-risk group, including T regulatory cells (Tregs), tumor-infiltrating lymphocyte (TIL), T helper cells, neutrophils, mast cells and immature dendritic cells (iDCs) (Fig 9D). There were four immune-related pathways significantly associated with the CS-ALPS, including type II Interferon (IFN) response, T cell co-inhibition, major histocompatibility complex (MHC) class I, and human leukocyte antigen (HLA) (Fig 9E). Furthermore, we discovered that the genes related to immune checkpoint were highly expressed in the low-risk group, particularly, such as *ICOS*, *BTNL2*, *TNFSF14*, *CD160*, *CD80*, *BTLA*, *TNFRSF25*, *TNFRSF14*, *CD27*, *CD200R1*, *CD28*, *CD48*, *TNFSF15*, *CD40LG*, *ADORA2A*, *IDO2* and *CTLA4*, etc. (Fig 9F).

## The CS-ALPS has a potential relationship with LUAD therapy

To explore whether the CS-ALPS was a guidance for LUAD therapy, we obtained the IC50 of some conventional chemotherapy drugs and targeted drugs as pharmacological data. The results showed that the IC50 of cisplatin, docetaxel, etoposide, paclitaxel, gemcitabine, erlotinib, crizotinib and afatinib was lower in high-risk group, meaning that the high-risk individuals were more sensitive to the majority of common chemotherapy drugs and targeted drugs (Fig 10A–10H). Therefore, the CS-ALPS could be used for predicting the efficacy of LUAD therapy and provided appropriate clinical guidance.

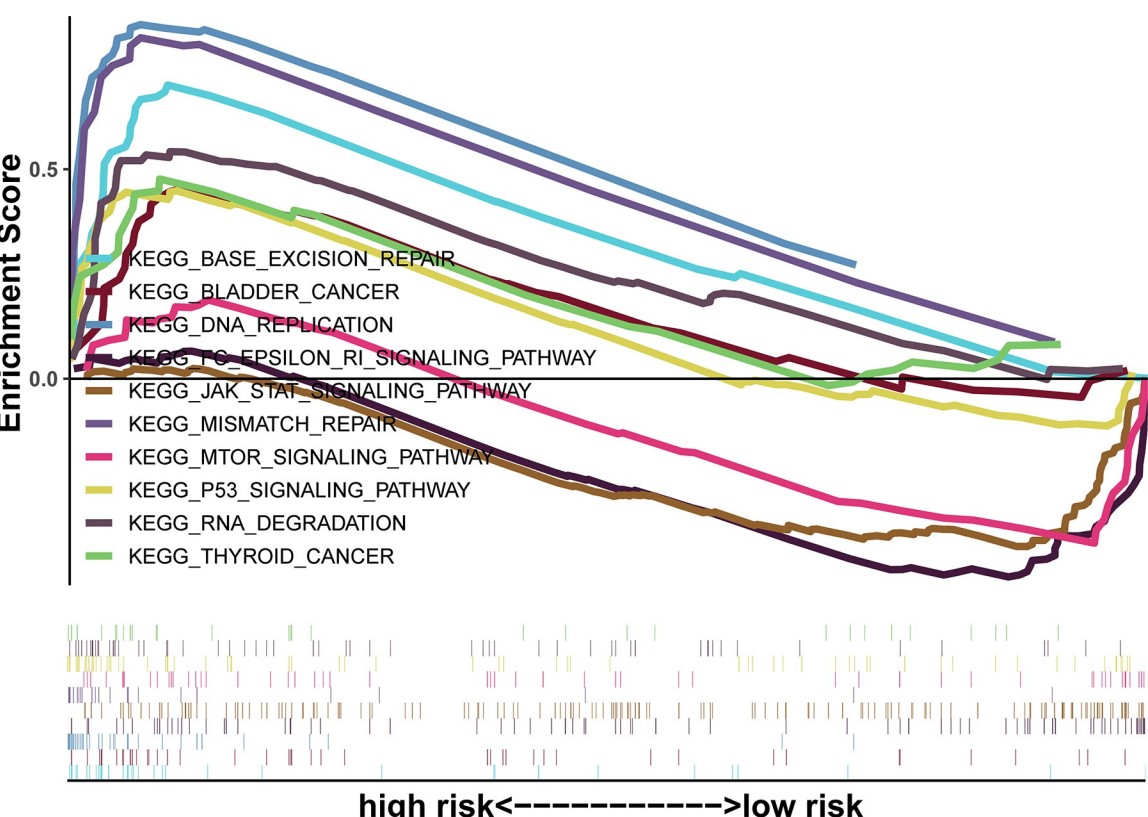

**Fig 8. The signaling pathways of significant enrichment according to GSEA.**

## Discussion

Currently, LUAD is the most epidemic pathological type that threatens non-smokers, and its incidence is persistently increasing year by year. Even though there are a host of treatments available, the prognosis of advanced LUAD remains very poor [24]. With the deepening research on tumor suppressor mechanism, the prevention of tumorigenesis and development by inducing tumor cellular senescence is gradually receiving attention. Cellular senescence is an anti-proliferative program that leads to permanent cell growth arrest [25] and protects cells from unnecessary damage [26]. In this study, we constructed a CS-ALPS for predicting the prognosis of LUAD. We obtained 62 cellular senescence-related DEGs, of which 5 genes were most closely associated with the other genes. Among them, *CDKN2A* encodes a nucleolar protein involved in triggering cell cycle arrest and apoptosis to suppress tumor growth [27]. Studies have shown that *CDKN2A* represses *MDM2*, a negative regulator of *p53*, thereby stabilizing *p53* and enabling it to generate a transcriptional response by relocating it to the nucleolus [28]. Liu et al. found that *CDKN2A* loss promoted the progression of lung cancer and was closely associated with poorer survival outcomes [29]. Cyclin dependent kinase 1 (*CDK1*) is one of the most important factors regulating the cell cycle, which is crucial for the time phase transitions of the cell cycle. Multiple bioinformatics analyses suggest that *CDK1* expressed highly in LUAD, negatively correlated to the OS of individuals with LUAD [30–32]. Li et al. showed that *miR-143-3p* could target and regulate *CDK1* and promote cancer cell apoptosis in LUAD A549 cell line [33].

In the enrichment analyses of 62 DEGs, we were pleasantly surprised to find that there was a significant enrichment in the *p53* pathway. A great deal of genes mediating cell cycle arrest can be regulated via *p53* signaling pathway [34]. The pathway is also involved in regulating

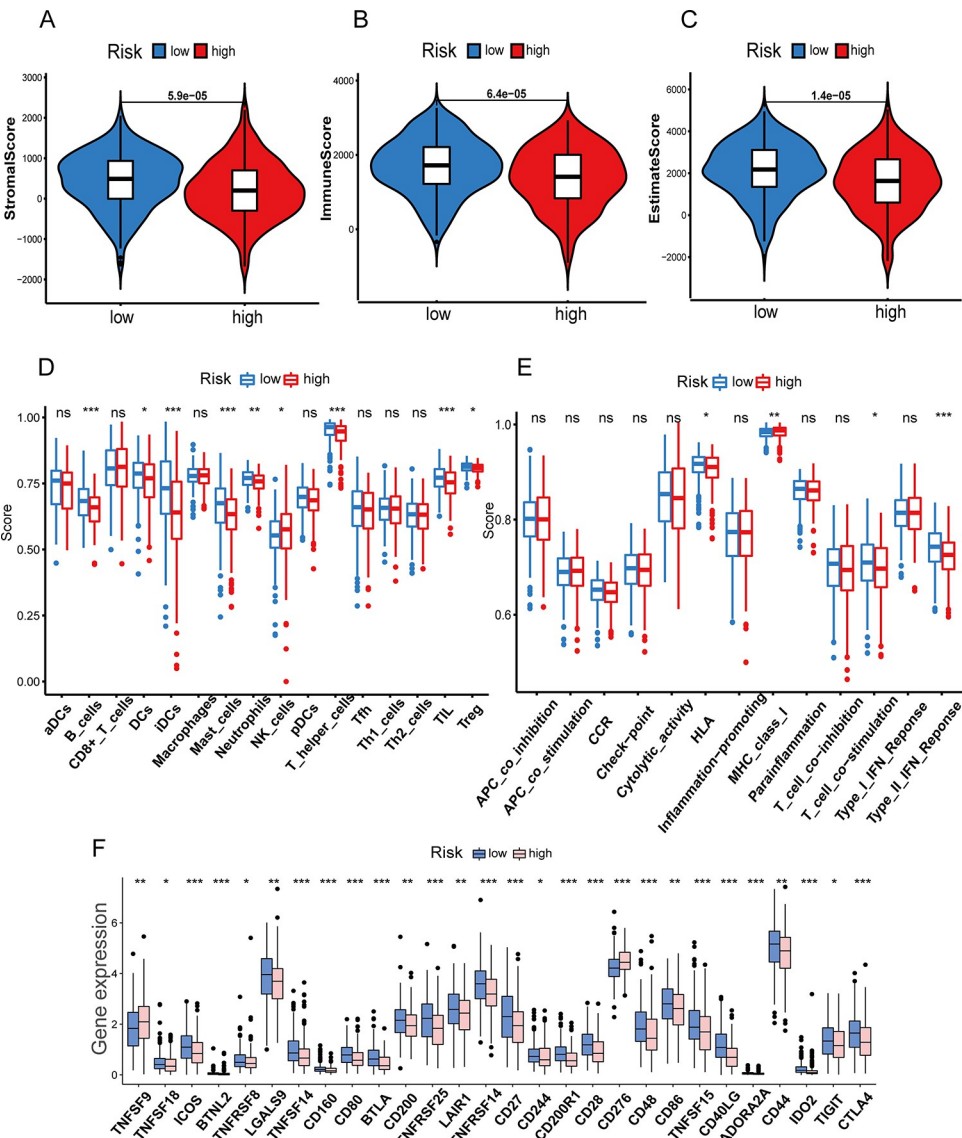

**Fig 9. Immune-related analyses.** (**A-C**) The violin plots of StromalScore, ImmuneScore, and ESTIMATEScore between two groups with high-and low-risk. (**D**, **E**) The enrichment scores of immune cells and immune-related pathways between two groups with high-and low-risk. (**F**) The expression of immune checkpoint between two groups with high-and low-risk. T, tumor; N, normal; M, metastasis; ns, non-significant; *p < 0.05; **p < 0.01; ***p < 0.001.

cellular senescence, which is consistent with our enrichment results [34,35]. Previous evidence has shown that the *p53* pathway can accelerate apoptosis and autophagy of LUAD cells [36,37]. Therefore, the enrichment result suggests that cellular senescence-related genes can inhibit the biological behavior of LUAD through the *p53* pathway.

The role of lncRNAs in LUAD has been verified by more and more studies. Roth et al. indicated that down-regulating the expression of *linc00673* could trigger the senescence of LUAD cells, thereby exerting an inhibitory effect on LUAD [38]. Tao et al. confirmed that lncRNA *MEG3* promoted etoposide-induced tumor cell senescence through the *miR-16-5p/VGLL4* axis [39]. Recently, Fang X et al. proposed a prognostic study of senescence-associated lncRNAs that tentatively elucidated the feasibility of constructing a prognostic model [40]. In

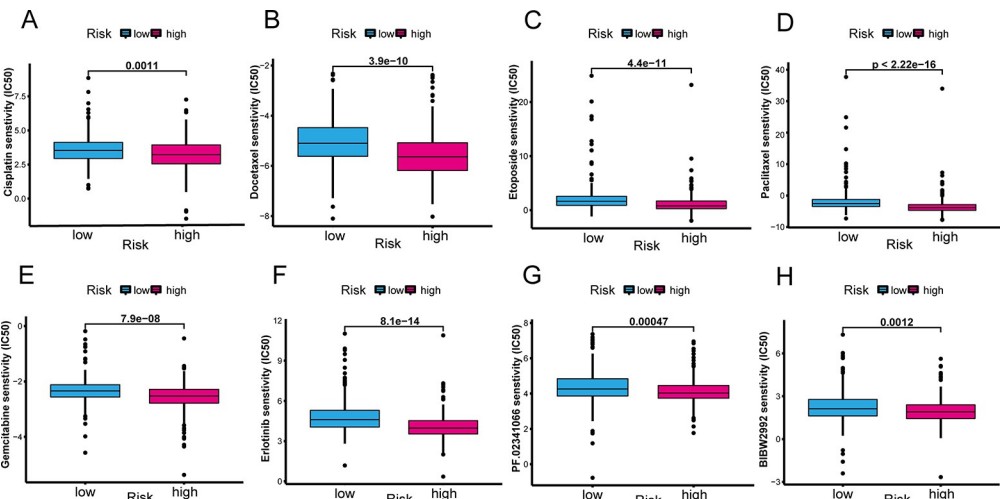

**Fig 10. The sensitivity analysis of common chemotherapeutic and targeted drugs.** (**A**-**E**) The chemotherapy drugs including Cisplatin, Docetaxel, Etoposide, Paclitaxel, Gemcitabine. (**F**-**H**) The targeted drugs including Erlotinib, PF.02341066 (Crizotinib) and BIBW2992 (Afatinib).

our study, we ultimately identified 6 cellular senescence-associated lncRNAs (*AF131215.5*, *GAS6-AS1*, *AC026355.1*, *GSEC*, *AL365181.2*, *C20orf197*) through multivariate COX regression analysis for building the predictive signature. We also found that 12 mRNAs (*ETS1*, *CBX7*, *SLC16A7*, *NTN4*, *PDZD2*, *KDM5B*, *WNT16*, *ASPH*, *IGFBP3*, *CHEK1*, *G6PD*, *GAPDH*) co-expressed with these 6 lncRNAs. Among them, high expression of NTN4 attenuates DNA damage-induced senescence in Glioblastoma multiforme (GBM) [41]. *PDZD2* is able to induce senescence in prostate cancer, breast cancer and liver cancer cells via the *p53* signaling pathway [42]. Knockdown of *GAPDH* can trigger cellular senescence in A549 cell line. Then, further experiments have found that *GAPDH* depletion may induce and accelerate cellular senescence of tumor through the *AMPK* network in the absence of DNA damage [43]. Besides, high expression of *G6PD* can also promote the growth of glioblastoma and inhibit its senescence [44].

In this research, LUAD patients were categorized as high and low risk according to the median risk score. The ROC curve suggested that the CS-ALPS had relatively outstanding predictive efficacy. Moreover, we randomly divided these patients into two internal validation cohorts and plotted ROC curves separately. The results suggested that they had the semblable predictive performance as the total cohort. The analysis of clinicopathological factors led to the conclusion that the CS-ALPS could be regarded as an independent prognostic indicator of LUAD patients.

GSEA showed that JAK/STAT pathway, mTOR pathway and FC epsilon RI pathway was enriched notably in the low-risk group. Thereinto, FC epsilon RI signaling pathway has many connections with immune function [45], while mTOR signaling pathway and JAK/STAT signaling pathway are closely associated with cancers. Many studies have indicated that the mTOR signaling pathway is one of the most common regulatory pathways which affected the occurrence, development and metastasis of LUAD by inducing cell apoptosis [46–48]. Rapamycin and temsirolimus as the mTOR inhibitors have already been used to treat NSCLC in various phases of clinical trials [49,50]. Currently, in accordance with more and more genetic and pharmacological evidence, mTOR activity is recognized as a main driver of cellular senescence [51]. Carroll B et al. found that *mTORC1* hyperactivity was an obvious characteristic of

cellular senescence, which could arrest the cell cycle [52]. Moreover, it was found that senescent cells produced pro-inflammatory cytokines to exacerbate aging-related tissue function decline by utilizing translational programs downstream of *mTORC1* [53,54]. The research showed that mTOR inhibition could not extend longevity of ATG-deficient worms, suggesting that the process of modulating lifespan through the mTOR signaling pathway might depend on autophagy mechanisms [55]. In addition, the rapamycin was found to extend lifespan in mice, further indicating the stimulative ageing effect of mTOR signaling pathway in the cellular senescence [56]. Based on the above studies, we believe that cellular senescence is a vital biological process in regulating LUAD via the mTOR signaling pathway. Furthermore, the JAK/STAT signaling pathway is also considered to be closely associated with biological behaviors of lung cancer, such as drug resistance [57]. Xu et al. verified that *JAK2* was positively linked to the proliferation and invasion of A549 cells [58]. There are also some evidences indicating that the JAK/STAT signaling pathway in senescent cells is more active than non-senescent cells and inhibition of the pathway can suppress the aging-related secretory phenotype and alleviate dysfunction in senescent cells [59].

The analysis of TIME showed StromalScore, ImmuneScore and ESTIMATEScore were significantly higher in low-risk groups. Therefore, we speculated that antitumor immunotherapy had a good therapeutic effect for the low-risk patients. Subsequently, ssGSEA also revealed a variety of immune cells, as well as some immune-related pathways, had higher immune scores in the group with low risk. Mayoux et al. demonstrated that DCs determined the treatment efficacy of PD-L1 immune checkpoint inhibitors [60]. A DC gene signature was strongly related to improvement of the OS in NSCLC patients through PD-L1 blockade treatment. Shikotra et al. discovered that the mast cells with different phenotypes were increased in the NSCLC, and the production of TNF-α by both mast cell phenotypes was particularly important for their interaction with other inmmune cells and the associated inhibition of tumor progression [61]. Researchers found that the Tregs with increased number and enhanced activity in the peripheral blood, metastatic lymph nodes and the tumors were positively related to the recurrence and metastasis of patients with NSCLC [62,63]. The immune checkpoint inhibitors have received increasing attention in recent years. We further analyzed the expression condition of genes related to immune checkpoint and revealed that the majority of checkpoints were more active in the low-risk group. These outcomes demonstrated that the patients at low risk were probably more responsive to immunotherapy, and the immune checkpoint inhibitors would improve the prognosis of these patients. Among these immune checkpoints, CTLA-4 inhibitors, as the effective drugs, were a milestone in the treatment of NSCLC [64]. Besides, *TIGIT* and *CD27* related immune checkpoints inhibitors have been under investigation in preclinical phase [65]. Lastly, our research showed that it was probably more advantageous for high-risk patients to accept the conventional chemotherapy drugs (cisplatin, docetaxel, etoposide, paclitaxel, and gemcitabine) and targeted therapy (erlotinib, crizotinib and afatinib).

To sum up, most of the results were helpful to explore a more precise and personalized therapeutic schedule for LUAD patients. Nevertheless, some limitations of our design are inevitable. Firstly, this study still requires an appropriate external dataset to validate the applicability of the CS-ALPS. Secondly, the molecular mechanism of cellular senescence-associated lncRNAs in LUAD need further experimental verification.

## Conclusion

In summary, the CS-ALPS can be regarded as an independent prognostic indicator of LUAD patients. It is helpful to explore the underlying molecular mechanism of lncRNAs associated

with cellular senescence in LUAD. From the perspective of clinical guidance, the CS-ALPS provides relatively scientific treatment strategies for LUAD patients. However, future experiments will be necessary to confirm its validity once more.

## Supporting information

**S1 Fig. A forest plot for 66 lncRNAs related to prognosis of LUAD.**
(PDF)

**S1 Table. 279 cellular senescence-related genes.**
(XLSX)

**S2 Table. 62 cellular senescence-related DEGs.**
(XLSX)

**S3 Table. 1081 cellular senescence-associated lncRNAs.**
(XLSX)

**S4 Table. Clinicopathological characteristics of LUAD patients from TCGA.**
(XLSX)

**S5 Table. 66 lncRNAs connected to the prognosis of LUAD.**
(XLSX)

**S6 Table. 29 cellular senescence-associated lncRNAs by LASSO.**
(XLSX)

**S7 Table. 6 cellular senescence-associated lncRNAs by multivariate Cox regression analysis.**
(XLSX)

**S1 File. 6 lncRNAs recheck and calculations of coding potential.**
(DOCX)

## Acknowledgments

We express gratitude to contributions from TCGA database and CellAge database.

## Author Contributions

**Conceptualization:** Anbang Liu, Xiaohuai Wang, Junhang Zhang, Yun Li.

**Data curation:** Anbang Liu, Xiaohuai Wang, Junhang Zhang, Yun Li.

**Formal analysis:** Anbang Liu, Dongqing Yan, Yin Yin, Yun Li.

**Funding acquisition:** Yun Li.

**Investigation:** Anbang Liu.

**Methodology:** Anbang Liu, Xiaohuai Wang.

**Validation:** Anbang Liu, Xiaohuai Wang.

**Visualization:** Hongjie Zheng, Gengqiu Liu, Junhang Zhang.

**Writing – original draft:** Anbang Liu, Xiaohuai Wang, Liu Hu.

**Writing – review & editing:** Anbang Liu, Xiaohuai Wang, Junhang Zhang, Yun Li.

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
