## [Decision Letter · Decision Letter 0]

14 Feb 2023

PONE-D-23-00324A novel cellular senescence-related lncRNAs predictive signature for the prognosis of lung adenocarcinomaPLOS ONE

Dear Dr. Li,

Thank you for submitting your manuscript to PLOS ONE. After careful consideration, we feel that it has merit but does not fully meet PLOS ONE’s publication criteria as it currently stands. Therefore, we invite you to submit a revised version of the manuscript that addresses the points raised during the review process.

 Please submit your revised manuscript by Mar 31 2023 11:59PM. If you will need more time than this to complete your revisions, please reply to this message or contact the journal office at plosone@plos.org. Please include the following items when submitting your revised manuscript:A rebuttal letter that responds to each point raised by the academic editor and reviewer(s). You should upload this letter as a separate file labeled 'Response to Reviewers'.A marked-up copy of your manuscript that highlights changes made to the original version. You should upload this as a separate file labeled 'Revised Manuscript with Track Changes'.An unmarked version of your revised paper without tracked changes. You should upload this as a separate file labeled 'Manuscript'.If applicable, we recommend that you deposit your laboratory protocols in protocols.io to enhance the reproducibility of your results. Protocols.io assigns your protocol its own identifier (DOI) so that it can be cited independently in the future. For instructions see: https://journals.plos.org/plosone/s/submission-guidelines#loc-laboratory-protocols. Additionally, PLOS ONE offers an option for publishing peer-reviewed Lab Protocol articles, which describe protocols hosted on protocols.io. Read more information on sharing protocols at https://plos.org/protocols?utm_medium=editorial-email&utm_source=authorletters&utm_campaign=protocols.

We look forward to receiving your revised manuscript.

Kind regards,

Divijendra Natha Reddy Sirigiri

Academic Editor

PLOS ONE

Journal Requirements:

2. Please note that PLOS ONE has specific guidelines on code sharing for submissions in which author-generated code underpins the findings in the manuscript. In these cases, all author-generated code must be made available without restrictions upon publication of the work. 

Please review our guidelines at https://journals.plos.org/plosone/s/materials-and-software-sharing#loc-sharing-code and ensure that your code is shared in a way that follows best practice and facilitates reproducibility and reuse.

"Include this sentence at the end of your statement: The funders had no role in study design, data collection and analysis, decision to publish, or preparation of the manuscript"

Reviewers' comments:

Reviewer's Responses to Questions

**Comments to the Author**

1. Is the manuscript technically sound, and do the data support the conclusions?

Reviewer #1: Yes

Reviewer #2: Yes

2. Has the statistical analysis been performed appropriately and rigorously? 

Reviewer #1: Yes

Reviewer #2: I Don't Know

3. Have the authors made all data underlying the findings in their manuscript fully available?

Reviewer #1: Yes

Reviewer #2: Yes

4. Is the manuscript presented in an intelligible fashion and written in standard English?

Reviewer #1: Yes

Reviewer #2: No

5. Review Comments to the Author

Reviewer #1: Gist/summary: The authors come up with an interesting story on

cellular senescence-related long noncoding RNAs (lncRNAs) in lung

adenocarcinoma (LUAD). To augment this, they predict prognostic values and check the TCGA datasets and perform a full construction of cellular

senescence-related lncRNAs predictive signature (CS-RLPS) which they claim to be the first of its kind in measuring the prognistic indicators.

Did the authors review and double check the bona fidelity of the lncRNAs from the databases such as NONCODE etc?

For example C20orf ids indee dmight have been reannotated and they could have been coding chunks by the tim ethe paper came out

Are there any coding potential calculations that the authors lookd into these lncRNAs esp looking them as guides ( cis or trans)?

The PPI and KEGG is well taken, but it would hav ebeen nice had authors shown the lncRNa-protein interatcions as well to check regulatome

The abstratc must be rewritten

The figures are of not high resolution, may be inserted with HR images

Minor but essential

Pl correct the following

L61: hepatocellular carcinoma is CLOSE to the WORST overall survival

L192: 39 days OLD

L198: lncRNAs WERE

L201; was SHOWN

l237: scatter plot ( tw different words ?)

L389 and L417: through THE

L431: risk groupS

L449: improve THE prognosis

L461: need ( remove s)

scores on a scale of 0-5 with 5 being the best

Language: 3.5

Novelty: 4

Brevity: 3.5

Scope and relevance: 4

Reviewer #2: 1) The study is good. However, needs improvements in terms of overall English language in sections of Abstract, Methods and results. Please invest enough time and a revision with main authors to bring out the best of the analysis in the text.

e.g. Abstract: Please modify this statement to fit the general format of Abstract (it appears to fit in methodology section)

“To illustrate the prognostic value of cellular senescence-related lncRNAs, The Cancer Genome Atlas (TCGA) database was accessible to download the RNA sequencing (RNA-seq) dataset, survival information, and associated clinicopathologic parameters of LUAD. The CellAge database provided the 279 genes involved in cellular senescence.” (The present statement is more like a Methodology section.)

2) An alternative title could be : "Predictive molecular signature consisting of lncRNA related to cellular senescence for the prognosis of lung adenocarcinoma ". This would be more appropriate,

3). Materials and methods section could start with explaining the flow diagram of analysis.

4) Results section – Each subtitle in this section should highlight the finding or outcome of analysis rather than the process or type of analysis! Please revise, this is very important to improve the quality of presentation of results and also to reach the quality of journal.

5) Figures: Please try to improve the resolution of all the figures. Texts in some of the figures is not readable at all.

6. PLOS authors have the option to publish the peer review history of their article (what does this mean?). If published, this will include your full peer review and any attached files.

Reviewer #1: No

Reviewer #2: No

---

## [Author Response · Author response to Decision Letter 0]

17 Mar 2023

Dear Editor:

Thank you for giving us the opportunity to revise our manuscript entitled “A novel cellular senescence-related lncRNAs predictive signature for the prognosis of lung adenocarcinoma”. We appreciate the reviewers’ constructive comments and their enthusiasm for our findings. Our point-to-point response to the issues raised by the referees is included. I hope that you would find out the revision is substantial and the reviewers’ concerns have been adequately addressed and agree that the manuscript is now acceptable for publication in PLOS ONE.

Sincerely,

Yun Li

liyun28@mail.sysu.edu.cn

Department of Thoracic Surgery, The Seventh Affiliated Hospital, Sun Yat-sen University, Shenzhen, China

Response to Editor’s comments

Authors: Thanks for your reminding. We have rechecked our manuscript and introduced minor changes to comply with the journal’s style requirements.

2. Please note that PLOS ONE has specific guidelines on code sharing for submissions in which author-generated code underpins the findings in the manuscript. In these cases, all author-generated code must be made available without restrictions upon publication of the work.

Authors: We are very happy to share our code if necessary.

3. About the financial disclosure.

Authors: The authors received no specific funding for this work.

4. In your Data Availability statement, you have not specified where the minimal data set underlying the results described in your manuscript can be found. PLOS defines a study's minimal data set as the underlying data used to reach the conclusions drawn in the manuscript and any additional data required to replicate the reported study findings in their entirety. All PLOS journals require that the minimal data set be made fully available.

Authors: According to your requirements, we will upload our study’s minimal underlying data set as supporting information when we re-submit our revised manuscript.

Authors: We have already had an ORCID iD and validated it in Editorial Manager. The ORCID iD of the corresponding author is 0000-0003-0949-6519.

6. About the references.

Authors: We appreciate greatly your important warnings and recheck our references. The 14th reference may involve academic misconduct. The 58th reference have been retracted, so we removed the two references and replaced them with relevant current references. Meanwhile, we added two references （the 16th, 17th）and revised the original contents accordingly.

Response to Reviewers’ comments

Reviewer #1:

1. Did the authors review and double check the fidelity of the lncRNAs from the databases such as NONCODE etc? For example, C20orf ids indeed might have been reannotated and they could have been coding chunks by the time the paper came out.

Authors: We have rechecked and ensured the fidelity of the 6 lncRNAs in signature from the LNCipedia database. We found the new annotation for the 6 lncRNAs and searched the relevant literature. For example, the alternative gene name of C20orf197 is lnc-CDH26-9, or LINC02910. The following paper came out also mentions the C20orf197. The others’ annotation and relevant paper was shown in our uploaded S1 File. 

Yao J, Chen X, Liu X, Li R, Zhou X, Qu Y. Characterization of a ferroptosis and iron-metabolism related lncRNA signature in lung adenocarcinoma. Cancer Cell Int. 2021;21(1):340. http://doi.org/10.1186/s12935-021-02027-2

2. Are there any coding potential calculations that the authors looked into these lncRNAs esp looking them as guides (cis or trans)?

Authors: We analyzed the coding potential of the 6 lncRNAs based on the LNCipedia database. All lncRNAs were determined to have no coding ability. The outcome of coding potential is shown in our uploaded S1 File.

3. The PPI and KEGG is well taken, but it would have been nice had authors shown the lncRNA-protein interactions as well to check regulation.

Authors: We showed the correlation analysis between the 6 lncRNAs and coding RNAs in Fig 3C. Regretfully, the interaction mechanism between lncRNAs and protein is so complicated that we are difficult to visualize them. 

4. The abstract must be rewritten.

Authors: We reviewed the section carefully and recognize the seriousness of the problem. We have rewritten and improved the overall structure and logicality of the abstract. The revised abstract will meet hopefully your requirements.

5. The figures are of not high resolution, may be inserted with HR images.

Authors: Thanks for your precious advice. We have revised and improved the resolution of all the figures.

6. Minor but essential, please correct the following:

L61: hepatocellular carcinoma is CLOSE to the WORST overall survival

L192: 39 days OLD

L198: lncRNAs WERE

L201; was SHOWN

l237: scatter plot (tw different words ?)

L389 and L417: through THE

L431: risk groupS

L449: improve THE prognosis

L461: need ( remove s)

Authors: We apologize for the mistakes of our manuscript and have revised the above mistakes one by one. We will be happy to edit the text further, based on helpful comments from the reviewers.

Reviewer #2:

1. The study is good. However, needs improvements in terms of overall English language in sections of Abstract, Methods and Results. Please invest enough time and a revision with main authors to bring out the best of the analysis in the text.

Authors: We apologize for the poor language of our manuscript. We have worked on the manuscript for a long time and carefully revised the content of Abstract, Methods and Result, including grammar, conciseness, readability, and logicality. Deletions and additions are both marked. We hope that the language level has been substantially improved. 

2. An alternative title could be: "Predictive molecular signature consisting of lncRNA related to cellular senescence for the prognosis of lung adenocarcinoma ". This would be more appropriate.

Authors: That’s a good idea. Thanks for your considerate suggestion. We think that the alternative title is more in line with the overall content of this manuscript. We decide to replace the original title with the new one. Moreover, we would like ask for your opinion. Whether we need add an “A” at the beginning of the new title. “A predictive molecular signature consisting of lncRNAs related to cellular senescence for the prognosis of lung adenocarcinoma”.

3. Materials and methods section could start with explaining the flow diagram of analysis.

Authors: We have adjusted the writing order of the Materials and methods according to the sequence of the flow diagram. It is very sorry that we did not fully understand your advice. Whether you require us to add an explanation of the flow diagram at the beginning of the Materials and methods section or not? Please let us know if it's necessary and we would like to follow up your suggestion. 

4. Results section – Each subtitle in this section should highlight the finding or outcome of analysis rather than the process or type of analysis! Please revise, this is very important to improve the quality of presentation of results and also to reach the quality of journal.

Authors: We appreciate greatly your pointing out the shortcomings and have already recognized the seriousness of the problem. We have carefully revised each subtitle to highlight the finding or outcome of analysis. 

5. Figures: Please try to improve the resolution of all the figures. Texts in some of the figures is not readable at all.

Authors: We are so sorry to bring poor visual experience to you. We have revised and improved the resolution of all the figures.

---

## [Decision Letter · Decision Letter 1]

11 May 2023

PONE-D-23-00324R1A predictive molecular signature consisting of lncRNAs related to cellular senescence for the prognosis of lung adenocarcinomaPLOS ONE

Dear Dr. Li,

Thank you for submitting your manuscript to PLOS ONE. After careful consideration, we feel that it has merit but does not fully meet PLOS ONE’s publication criteria as it currently stands. Therefore, we invite you to submit a revised version of the manuscript that addresses the points raised during the review process.

As per my understanding from reviewers comments, there are still some issues to be addressed. Please address these minor issues. 

We look forward to receiving your revised manuscript.

Kind regards,

Divijendra Natha Reddy Sirigiri

Academic Editor

PLOS ONE

Journal Requirements:

Reviewers' comments:

Reviewer's Responses to Questions

**Comments to the Author**

1. If the authors have adequately addressed your comments raised in a previous round of review and you feel that this manuscript is now acceptable for publication, you may indicate that here to bypass the “Comments to the Author” section, enter your conflict of interest statement in the “Confidential to Editor” section, and submit your "Accept" recommendation.

Reviewer #1: All comments have been addressed

Reviewer #2: All comments have been addressed

2. Is the manuscript technically sound, and do the data support the conclusions?

Reviewer #1: Yes

Reviewer #2: Yes

3. Has the statistical analysis been performed appropriately and rigorously? 

Reviewer #1: Yes

Reviewer #2: Yes

4. Have the authors made all data underlying the findings in their manuscript fully available?

Reviewer #1: Yes

Reviewer #2: Yes

5. Is the manuscript presented in an intelligible fashion and written in standard English?

Reviewer #1: (No Response)

Reviewer #2: Yes

6. Review Comments to the Author

Reviewer #1: I am satisfied with all the changes rendered. Thank you

Pl double check the language for brevity

The figure smay be of HR

Reviewer #2: Please address the following minor issues and see if the suggestions would help to further improve the MS.

1) Similar article has appeared in Frontiers Genetics2022:

A novel senescence-related lncRNA signature that predicts prognosis and the tumor microenvironment in patients with lung adenocarcinoma.

Fang X, Huang E, Xie X, Yang K, Wang S, Huang X, Song M. Front Genet. 2022 Nov 3;13:951311. doi: 10.3389/fgene.2022.951311. eCollection 2022.

Please check it and if it’s needs to be quoted and if it has any of the lncRNA identified by your study are also outcome of this study, which might strengthen your claims. One is seen - AC026355.1.

2) Abstract – line one, and there is still scope to improve the abstract furthermore with language.

# line 239-240 –pathogenesis and clinical factors.

#447 -”Independently” -> independent

3) If all the lncRNA that are found in this study are already shown (demonstrated) in other studies functionally - that they are involved in cellular senescence, then only it would be appropriate to quote these lnRNAs as “cellular-senescence-related” otherwise it would be more appropriate to quote them as “Cellular senescence-associated-lncRNA”, and in entire text.

4) line 67-68 - it's contrary to reports stating that - cellular senescence hinders or delays tumorigenesis!! Cellular senescence is a mechanism by which organism delays the onset of tumorigenesis.

5) Result section:

A) Line 247-249 - “Risk score = (-0.270×AC026355.1 expression, 0.231×AL365181.2 expression, -0.616×AF131215.5 expression, -0.580×C20orf197 expression, -0.627×GAS6-AS1 expression, 0.505×GSEC expression)”.

# in this formula: What is meant by ","(a comma)? Does it mean addition (+) ? And ‘–’ Does this hyphen mean “minus/subtraction”?

B) Please check in the text attached pdf file with marking and comment.

224 / 257 /295 -296 / 330 / 346-347 and at some other points.

#414 - Therefore the enrichment result suggests that the…

# 468 – indicating

C) Result subtitles – It would be great if the subtitles would also contain or highlight the biological outcomes (in one line- or in a concise statement) explained under these subtitles.

7. PLOS authors have the option to publish the peer review history of their article (what does this mean?). If published, this will include your full peer review and any attached files.

Reviewer #1: **Yes: **Prashanth N Suravajhala

Reviewer #2: **Yes: **Dr. Avinash Arvind Rasalkar

---

## [Author Response · Author response to Decision Letter 1]

19 May 2023

Dear Editor:

 Thank you for giving us the opportunity to revise our manuscript entitled “A predictive molecular signature consisting of lncRNAs associated with cellular senescence for the prognosis of lung adenocarcinoma”. We appreciate the referee’s constructive comments and their enthusiasm for our findings. Our point-to-point response to the issues raised by the referees is included. I hope that you would find out the revision is substantial and the referees’ concerns have been adequately addressed and agree that the manuscript is now acceptable for publication in PLOS ONE.

 Sincerely,

 Yun Li

liyun28@mail.sysu.edu.cn

 Department of Thoracic Surgery, The Seventh Affiliated Hospital, Sun Yat-sen University, Shenzhen, China

Response to Reviewers’ comments

Reviewer #1:

1.I am satisfied with all the changes rendered. Thank you. Pl double check the language for brevity. The figure smay be of HR.

Authors: Thanks for your review. We have checked the language and made much revision again to ensure its brevity and correctness. Meanwhile, the resolution of all the figures has been improved.

Reviewer #2:

1.Similar article has appeared in Frontiers Genetics2022:

A novel senescence-related lncRNA signature that predicts prognosis and the tumor microenvironment in patients with lung adenocarcinoma.

Fang X, Huang E, Xie X, Yang K, Wang S, Huang X, Song M. Front Genet. 2022 Nov 3; 13:951311. doi: 10.3389/fgene.2022.951311. eCollection 2022.

Please check it and if it’s needs to be quoted and if it has any of the lncRNA identified by your study are also outcome of this study, which might strengthen your claims. One is seen - AC026355.1.

Authors: That’s a good idea. We think that the article is valuable and we have quoted it as the 40th reference.

[38] Fang X, Huang E, Xie X, Yang K, Wang S, Huang X, et al. A novel senescence-related lncRNA signature that predicts prognosis and the tumor microenvironment in patients with lung adenocarcinoma. Front Genet. 2022; 13:951311. http://doi.org/10.3389/fgene.2022.951311

2.Abstract – line one, and there is still scope to improve the abstract furthermore with language.

# line 239-240 –pathogenesis and clinical factors.

#447 - “Independently” -> independent.

Authors: Thanks for your point. We think that the “clinicopathological” is appropriate, because we found the usage in the following article.

Chen M, Nie Z, Li Y, Gao Y, Wen X, Cao H, et al. A New Ferroptosis-Related lncRNA Signature Predicts the Prognosis of Bladder Cancer Patients. Front Cell Dev Biol. 2021; 9:699804. http://doi.org/10.3389/fcell.2021.699804

According to your comments, we have addressed the rest of problems.

3.If all the lncRNA that are found in this study are already shown (demonstrated) in other studies functionally - that they are involved in cellular senescence, then only it would be appropriate to quote these lnRNAs as “cellular-senescence-related” otherwise it would be more appropriate to quote them as “Cellular senescence-associated-lncRNA”, and in entire text.

Authors: We think that raised point is precise. All the “cellular senescence-related lncRNAs” have been replaced with “Cellular senescence-associated lncRNA”. 

4.line 67-68 - it's contrary to reports stating that - cellular senescence hinders or delays tumorigenesis!! Cellular senescence is a mechanism by which organism delays the onset of tumorigenesis.

Authors: We appreciate greatly you for pointing out the shortcomings and have recognized the seriousness of the problem. We have carefully revised it and changed the 3rd and 4th references.

[3] Lee S, Lee JS. Cellular senescence: a promising strategy for cancer therapy. Bmb Rep. 2019;52(1):35-41. http://doi.org/10.5483/BMBRep.2019.52.1.294

[4] Demirci D, Dayanc B, Mazi FA, Senturk S. The Jekyll and Hyde of Cellular Senescence in Cancer. Cells-Basel. 2021;10(2). http://doi.org/10.3390/cells10020208

5.A) Line 247-249 -“Risk score = (-0.270×AC026355.1 expression, 0.231×AL365181.2 expression, -0.616×AF131215.5 expression, -0.580×C20orf197 expression, -0.627×GAS6-AS1 expression, 0.505×GSEC expression)”.

# in this formula: What is meant by ","(a comma)? Does it mean addition (+) ? And ‘–’ Does this hyphen mean “minus/subtraction”?

Authors: We are so sorry for this mistake. ","(a comma) means addition (+). ‘–’ means negative correlation coefficient.

Risk score = (-0.270×AC026355.1 expression) + (0.231×AL365181.2 expression) + (-0.616×AF131215.5 expression) + (-0.580×C20orf197 expression) + (-0.627×GAS6-AS1 expression) + (0.505×GSEC expression).

B)Please check in the text attached pdf file with marking and comment.224 / 257 /295 -296 / 330 / 346-347 and at some other points. #414 - Therefore the enrichment result suggests that the… # 468 – indicating.

Authors: According to your comments, we have revised these issues one by one.

C)Result subtitles – It would be great if the subtitles would also contain or highlight the biological outcomes (in one line- or in a concise statement) explained under these subtitles.

Authors: Thanks for your suggestions. We have revised these subtitles to highlight the biological outcomes.

---

## [Editor Report · Decision Letter 2]

31 May 2023

A predictive molecular signature consisting of lncRNAs associated with cellular senescence for the prognosis of lung adenocarcinoma

PONE-D-23-00324R2

Dear Dr. Li,

We’re pleased to inform you that your manuscript has been judged scientifically suitable for publication and will be formally accepted for publication once it meets all outstanding technical requirements.

Kind regards,

Divijendra Natha Reddy Sirigiri

Academic Editor

PLOS ONE
---

## [Editor Report · Acceptance letter]

12 Jun 2023

PONE-D-23-00324R2 

A predictive molecular signature consisting of lncRNAs associated with cellular senescence for the prognosis of lung adenocarcinoma 

Dear Dr. Li:

I'm pleased to inform you that your manuscript has been deemed suitable for publication in PLOS ONE. Congratulations! Your manuscript is now with our production department. 

Kind regards, 

on behalf of

Dr. Divijendra Natha Reddy Sirigiri 

Academic Editor

PLOS ONE